# The Relationship between Trauma and Attachment in Burundi’s School-Aged Children

**DOI:** 10.3390/brainsci13040666

**Published:** 2023-04-15

**Authors:** Jean Bosco Ndayishimiye, Barry H. Schneider, Léandre Simbananiye, Thierry Baubet

**Affiliations:** 1Doctoral School of Burundi, Bujumbura 1550, Burundi; ndaye2014@gmail.com; 2Boston College, Department of Psychology and Neuroscience, University of Ottawa, Ottawa, ON K1N 9B2, Canada; schneibc@bc.edu; 3Centre de Recherche et D’intervention Pour le Développement Individuel et Communautaire (CRIDIS), Université du Burundi, Bujumbura 1550, Burundi; 4Department of Child and Adolescent Psychiatry, Hôpital Avicenne, Assistance Publique—Hôpitaux de Paris, F-93000 Bobigny, France; thierry.baubet@aphp.fr; 5Unité Transversale de Psychogénèse et Psychopathologie (UTRPP—EA 4403), Université Sorbonne Paris Nord, F-93430 Villetaneuse, France; 6Centre National de Ressources et de Résilience (CN2R), F-59000 Lille, France

**Keywords:** relational trauma, children, adolescents, trauma, insecure attachment, secure attachment, traumatic experience

## Abstract

The exposure of children and adolescents to trauma is one of the most important public health challenges. These childhood experiences play a role in children’s attachment patterns with their parents and peers. The objective of this study was to examine the relationship between exposure to trauma and the degree of attachment representations in school-aged children in Burundi. One hundred thirteen vulnerable children aged 7 to 12 years were recruited and referred by their teachers. We used an event list including the post-traumatic reaction index to measure their exposure to traumatic events and the People in My Life instrument to measure attachment representations. The results revealed that the children had experienced or witnessed at least one traumatic event. The results indicated that secure attachment representations were highest among children with their parents and lowest among children with their peers. The relationship between trauma experiences and children’s attachment representations was significant with their parents and with their peers. Children’s attachment representations with their parents and peers predicted their traumatic experiences. Future research should focus on how attachment relationships can facilitate counselors and clinicians in providing preventive psycho-education to adults and children to develop healthier functioning, through better knowledge of the complex interplay between traumas.

## 1. Introduction

The exposure of children and adolescents to trauma is one of the most important public health challenges. These childhood experiences play a role in children’s attachment patterns with their parents and peers. The term trauma is derived from the Greek word for “injury”, which has been used for centuries in medicine to refer to “an injury to living tissue caused by an extrinsic agent”. It is defined, on the one hand, as being associated with a single, time-limited traumatic event (illness, divorce, natural disaster, etc.); on the other hand, it follows repeated long-term exposure to a traumatic event (sexual abuse, neglect, maltreatment, etc.). Exposure to trauma is a common experience for children and adolescents [1]. Childhood trauma, including abuse and neglect, is probably the most important public health challenge [2]. The Substance Abuse and Mental Health Services Administration’s statistical study shows that more than two-thirds of children reported experiencing at least one traumatic event by age 16 [3]. Physical trauma [4] and emotional neglect maltreatment and abuse of a child reflect a lack of respect for the needs of the child, and often a relational context marked by fragility or failures in the bonds of attachment [5]. Physical, sexual, and emotional abuse can lead to relational trauma, as can neglect, parental abandonment, and inconsistent prenatal education or care that includes episodes of neglect and abuse [6]. Experiences of relational trauma at a young age can affect childhood relationships and lead to difficulties in adult relationships [7]. From then on, early childhood is marked by a dependence on and the permeability of the custodial relationship; relational trauma can create an “insoluble paradox” [8], in which the love object is also the source of a threat to the infant’s psychological integrity. In the face of this situation, the child displays attachment behavior that allows them to achieve or maintain proximity to another identified individual who is conceived as being better able to cope with the world. This is most evident whenever the individual is frightened, tired, or ill and is soothed by comfort and care [9].

The attachment bond consists of an individual using another as a “safe base” from which to explore and master the environment in times of security and as a “haven” in times of stress or danger [10]. Secure parental attachment in children allows them to see their parents as confidants, while an emotionally kind father is a source of security for his children [11]. The first relationship the child establishes with their mother determines the type of attachment between the mother and child. A healthy parent-child attachment has positive impacts. These positive impacts have long-term effects on the child’s developmental outcomes. Conversely, repeated rejection, emotional inconsistency, and carelessness on the part of the primary caregiver toward the child are factors that lead to maladjustment in attachment development [12]. Bowlby’s attachment theory states that when a child’s emotional needs are not met by a caregiver, an insecure attachment may result [13].

Studies in Burundi [14,15] show that children in street situations and victims of trauma including sexual abuse, physical abuse, and emotional abuse have difficulties related to post-traumatic stress disorder. No studies have been performed on traumatized Burundian school-aged children. With this study, we wanted to examine the relationship between the levels of exposure to traumatic experiences and the degree of attachment representations of Burundian school-aged children with their parents and their peers. We hypothesized that levels of exposure to traumatic experiences predict the degree of attachment representation children have with their parents and peers. We expect that the variation in children’s degree of attachment is explained by the level of exposure to childhood traumatic experiences. Finally, we described the prevalence of the rate of exposure to traumatic experiences in Burundian school-aged children.

## 2. Materials and Methods

### 2.1. Correlational Design

Therefore, the design of our study is correlational and includes more sophisticated statistics such as logistic regression and multiple regression. The scientific method adopted for the quantitative elaboration of our research is a correlational type for a mainly cross-sectional study of childhood trauma and the degree of attachment representations in school-aged children. In recognition of the fact that we cannot say with certainty that one has a causal effect on the other, theoretically, we propose that the predictor variable precedes the criterion variable. Our sample population is naturally based on the actual cases of children who are victims of the identified early traumas. However, since our study assesses elements of the subject’s personal history, such as traumatic experiences, this type of variable can be considered a predictor of later behaviors.

In this study, the independent variable (exposure to traumatic events) is called the predictor, and the dependent variable (the degree of attachment representations) is called the criterion. From these two variables, the analysis of the results by simple linear regression will make it possible to explain the relationships between the variables using a right-hand equation. Unlike multiple linear regression, simple linear regression does not add other independent variables to the equation. It allows us to know if the dependent variable can predict the independent variable. Thus, for this regression, here is the hypothesis:

**H0**. 
*The degree of attachment representations is not related to early relational experiences before the age of 6.*


**H1**.The degree of attachment representations is related to early relational experiences before the age of 6.

### 2.2. Participants

Our survey population is drawn from a list of 500 students, 300 of whom are girls and 200 of whom are boys, all of whom are vulnerable, drawn up by the Maranvya communal administration for the 2021–2022 school year, from which we sampled according to the child’s gender. Since the student population is divided into two distinct strata, boys and girls, the appropriate sampling method is stratified random sampling. A stratified random sample is a sample that combines several random samples from separate groups within the population. The sample size of each group reflects the proportion of that group, or stratum, in the population [16]. Applying this to the student population, 300 out of 500 students are girls. Therefore, the proportion of girls is 300/500 = 0.6, which in percentage terms is 0.6 × 100% = 60%. Given the feasibility of our study, we decided that our population size is 150 students. This means that to reflect the proportions of boys and girls in the population, 60% of our sample must be girls. Therefore, we selected 90 girls and 60 boys for the study.

Thus, our sample consisted of 150 children, including 90 girls and 60 boys, all recognized as vulnerable by the communal administration and all adolescents aged between 7 and 12 years who were attending the three basic schools built in the vicinity of a “palessie” site for returnees and survivors of war violence in Burundi in the province of Bujumbura, in the commune of Mutimbuzi, and specifically in the Maramvya zone, at the time of data collection. We excluded 10 participants (6 girls and 4 boys) from the analyses because they were the only participants who did not live at the site and/or in the vicinity of the site, and we wanted to avoid bias in the sample; we also excluded 20 children (12 female and 8 male) because of concerns about the reliability of the data during the project, and finally excluded 7 female children from the study because of a lack of transparency about their housing status that made them ineligible for inclusion during the previous 4 months. The inclusion criteria were to be between 7 and 12 years of age, to be able to understand and respond independently to the two blindly translated Kirundi questionnaires, and to have no serious mental illness or developmental delay. Another criterion for exclusion from the sample was that the children did not have an intellectual disability or an autism spectrum disorder, which are associated with significant difficulties in middle school. Due to these adjustments, analyses including the degree of attachment representations were conducted with *N* = 113 (72 girls and 41 boys). The children were recruited from three basic schools in the Maramvya zone, Commune Mutimbuzi, Bujumbura province: Maranvya IV Fundamental School (*n* = 31), Buhomba Fundamental School (*n* = 55), and Village of Hope Burundi (*n* = 37), and were referred by their class teachers and school authorities who recognized them as vulnerable.

### 2.3. Procedure

Before starting, it was important to have the agreement of the director of the institution to find a way to inform and obtain the consent of the children who wanted to participate and to know those who did not want to participate. When we met with each child, we read the consent form, which states that they have the choice to participate, that they are not obligated to do so, and that they can withdraw the data that we have collected.

### 2.4. Data Analysis

Statistical analyses were performed using the IBM SPSS 26.0 Statistics program. The normal distribution of all relevant variables was tested using the Kolmogorov–Smirnov test. The significant results (*p* > 0.05) indicate that the distributions are normal. The results of the skewness test, the kurtosis test, and the *p*-*p* graphs allowed us to decide that the variable could be considered as normally distributed. The results (absolute values below 1.96 for skewness and kurtosis in the z-transformed variables) were considered normally distributed when checking for anomalies in the *p*-*p* plots [17]. All z-transformed total scores met the stated criteria and were, therefore, treated as normally distributed. The means are considered significantly different when the *p*-value is less than 0.05 (the same applies to the *p*-value of the chi-squared). Following the confirmation of the normality test, we conducted two basic regression linear regression analyses to examine the relationship between the degree of traumatic exposure and the children’s levels of attachment to their parents, on the one hand, and their levels of attachment to their peers, on the other hand.

### 2.5. Survey Measures

The next section describes the different measurement instruments that were used in this study. Once the various consents were signed, we administered the questionnaires to the children. The English questionnaires for children have translated well ahead of time into Kirundi, were discussed by a bilingual research team to confirm the semantic equivalence of items between versions, and were reviewed by the survey team to ensure contextual appropriateness.

Our population of interest is children between the ages of 7 and 12, with children who are being cared for after their victimization and children who are victims of trauma and continue to live in the environment where the traumatic event occurred.

#### 2.5.1. Demographic Questionnaire

Basic information about the child’s gender and age was recorded on lists provided by the principals of the schools involved and was collected before the actual survey.

#### 2.5.2. Measurement Instrument

All instruments were blindly translated and back-translated from a validated English or French version into Kirundi. Differences in meaning were discussed with the two translators and a team of local psychologists until a consensus was reached.

The original instruments were translated from English into Kirundi by the first author and back-translated into English by the second author to verify the accuracy of the translation. A series of back translations were employed to verify the accuracy of the translations. The first author, a native speaker of Kirundi, prepared the initial translation and compared it to a back translation prepared by the second author, a native speaker of English. Any discrepancies were discussed, and the questionnaires were revised and back-translated in a few cases where necessary. The questionnaires were administered in the context of the teaching language of each school in Kirundi.

#### 2.5.3. Exposure to Traumatic Events

Exposure to traumatic events for a lifetime was assessed through the list of included events from the University of California Los Angeles (UCLA) PTSD Reaction Index for Children and Adolescents [18]. This checklist includes 14 different events, covering both war and non-war traumatic events (e.g., natural disasters, accidents, natural disasters, accidents, sexual violence, physical abuse, physical neglect, etc.). Responses are coded dichotomously (yes = 1, no = 0), with a possible total score ranging from 0 to 14. Thus, an overall PTSD severity score can be calculated by adding the symptom scores, resulting in a maximum possible score of 68. A diagnosis of PTSD was made if DSM-IV criteria were met, including impairment of children’s daily functioning in response to traumatic stress. The UCLA PTSD Index has good psychometric properties and has been successfully used and validated in non-Western and African settings [19]. Inter-rater reliability was assessed by independently assessing the same child in parallel, that is when both investigators were present. The intraclass correlation of 0.99 (*p* < 0.001) indicated strong agreement between investigators.

#### 2.5.4. People in My Life Questionnaire

Attachment measures are available for early childhood and preschool [20]. However, this is not the case for the rest of the school period. This means that during this period, attachment behaviors are no longer recognized primarily by the search for closeness to parents. In the same way, although very relevant for observing attachment behaviors in early childhood, the separation–reunion procedure is not appropriate for children after the beginning of school age because they experience daily separations from their parents. Thus, attachment at school age is not assessed by behavior but by attachment representations. The attachment relationships of children and young adolescents with their parents and peers were assessed using the People in My Life parent and peer scales. This self-report attachment instrument assesses children’s relationships with their parents, peers, school, and neighborhood. It is designed for children and young adolescents and was developed from a sample of 320 students with an average age of 11 years. Each item is rated on a four-point scale.

The original version of the People in My Life questionnaire [21] is a downward extension of the Parent and Peer Attachment Inventory [22], which was originally developed through factor analysis with a sample of university students to tap into the behavioral elements of adolescent attachment and affective tonic cognitive expectations, suggesting internal working models of attachment to parents and close friends. It has two scales (the 21-item Parent Attachment Scale and the 26-item Peer Attachment Scale), each of which includes three dimensions: (1) trust (e.g., “My parents respect my feelings”; “My friends accept me as I am”); (2) communication (e.g., “I talk to my parents when I have a problem”; “I share my thoughts and feelings with my friends”); and (3) alienation (e.g., “I feel angry with my parents”; “I get angry easily with my friends”). The Peer Scale also includes a delinquency subscale (3 items) that assesses children’s agreement with their friends’ delinquent behavior (e.g., “If one of my friends asked me to skip school, I would do it”).

The PIML items are answered using a four-item Likert scale ranging from 1 (rarely or never true) to 4 (almost always or always true), and each subscale score is made up of the sum of the items, with higher scores indicating higher levels of trust, communication, alienation, and delinquency. In both scales (parent attachment and peer attachment), it is also possible to obtain an overall attachment score by summing all items, after reversing the coding of the alienation subscale items. The parenting scale consists of 21 items: 10 assess trust, 5 assess communication, and 6 assess alienation. The reliability coefficient for the parent scale is 0.869, and the alpha coefficients for the subscales range from 0.65 for alienation to 0.87 for trust. The peer scale consists of 24 items: 12 assess trust, 5 assess communication, and 7 assess alienation. The items in each scale are summed and can be divided by the number of items to obtain an average between 1 and 4. Adding the trust and communication scores and subtracting the alienation score yields the total parent scale, with scores that can range from −2 to 7. Adding the trust and communication scores and subtracting the alienation and scores yields the total peer scale, with scores that can range from −6 to 6. 

The results of the reliability test of the questionnaire translated into Kirundi reveal a significant level of internal consistency between the dimensions of parental attachment relationships (with a Cronbach’s alpha index of 0.869) and the dimensions of children’s attachment relationships with their peers (with a Cronbach’s alpha index of 0.769).

### 2.6. Ethical Precautions

The implementation of this thesis will not jeopardize the subjects’ physical or psychological well-being; however because the subject of our research involves traumas and the consequences on mental health pathologies in victims, the letters of information and consent must be comprehensive for everyone, while also considering the impact that this may have on people. The students and their teachers had the option to stop participating at any time without having to justify themselves, understanding that the information gathered would be kept private and anonymized within the study. The signed consent of each teacher as well as each student’s consent were obtained. At the meeting of the parents and the educational authorities with the researchers, it was noted that the parents and/or tutors had, in their own right, given the children permission to participate in the study.

## 3. Results

### 3.1. Descriptive Statistics

All children included in the analysis were male (36.3%) and female (63.7%) between 7 and 12 years of age (M = 9.4 and SD = 2.035). The children’s ages at the time of exposure to the events were a minimum of 4 years and a maximum of 7 years (M = 5.54 and SD = 0.721). On average, children had experienced an M = 1.37 (SD = 0.537) range of traumatic events over their lifetime. Figure 1 provides an overview of the rates of traumatic event exposure.

### 3.2. Rate of Exposure to Traumatic Events

All of the children surveyed described and reported having experienced or witnessed at least one traumatic event, as defined by PTSD criterion A1 (Figure 1). The results of this study (Table 1), using the UCLA PTSTD R-5 as the information collection tool in its trauma self-report section, reveal that the children surveyed had experienced or witnessed at least one traumatic event: either an unintentional trauma such as being in a bad accident (accident, 33.6%), being in a place where a war was going on around them (war, 56.4%), or being in a natural disaster (disaster, 56.6%) or an intentional trauma such as having experienced events such as being hit or punched (domestic violence victim, 56.4%); seeing a family member be hit, punched, or kicked very hard at home (domestic violence witness, 58.4%); being beaten up, shot, or threatened with serious injury in their school (community violence victims, 63.6%); seeing a dead body (witnessed a dead body, 62.8%); having someone touch their private parts when they did not want to (sexual assault, 6.2%); seeing someone who was beaten, shot, or killed (witnessed community violence, 64.6%); seeing or hearing about the violent death or serious injury of a loved one or friend (witnessed death or injury of loved one, 61.9%); undergoing painful or frightening medical treatment while very ill or seriously injured (serious psychological condition, 69.9%); being sexually abused (60.2%); or experiencing a deceased loved one (61.1%). The levels of exposure to traumatic events are very high at 42.28%, and children who report no exposure to trauma are at 37.17%, while mild and moderate levels are at 7.08% and 13.27%, respectively (Figure 2).

### 3.3. Degree of Attachment Representations of School-Aged Children

The degree of attachment representations was evaluated using a questionnaire that showed its metric qualities with a Cronbach’s alpha index of 0.890. However, we looked to see if all the items were in the same direction, and we proceeded to eliminate certain items from the questionnaire that could compromise the structure; the results indicated a good Cronbach’s alpha index of 0.895. This allowed us to see that there were no longer any items that disrupted the structure. The results of this work (Table 2) show that children with higher levels of insecure and disorganized attachment representations with peers (*n* = 73 or 72.30%) are more frequent than children with secure attachment representations (*n* = 40 or 27.69%). These same results show that children’s attachment representations with their parents are more secure (*n* = 86 or 76.10%) than insecure (*n* = 27 or 23.89%). The results of the parametric nominal test reveal that the distributions of the scores for the degree of attachment of the children with their parents are not the same in the different categories of the living environment of the children (with the Kruskall–Wallis test’s *p*-value < 0.005 = 0.012 for the degree of representations of attachment of the children with their parents and *p*-value = 0.000 < 0.005 for the representation of attachment of the children with their peers). The distribution of the degree of attachment representations for children with their parents is asymptotic (Figure 3a). The same is true for the unequal distributions of attachment representation scores with peers (Figure 3b). Finally, the normality tests of Kolmogorov–Smirnov, and Shapiro–Wilk (*p*-value = 0.000) show that the distribution of the scores of the degree of representation of children with their parents and their peers is normal (Figure 2c).

### 3.4. Relationship between Traumatic Event Exposure and Children’s Attachment Representations with Their Parents and Peers

The correlation between attachment with parents and attachment with peers is significant but quite low, as it is never higher than 0.30 (Pearson’s correlation test: *p*-value = 0.280). Therefore, it is unsustainable to combine them to make an overall attachment score, whatever the articles on the instrument may say. We, therefore, sought to study the relationship between exposure to events and the degree of attachment representations with parents and with peers. To test the relationship between children’s exposure to events and their degree of attachment representations with parents, a multiple regression analysis was performed. On the one hand, the results of the linear regression test (R = 0.230, F = 6.194, ddl = 1, and *p*-value = 0.014) indicate that exposure to traumatic events is weakly significant to the degree of attachment representations of the children surveyed; it, therefore, follows that exposure to traumatic events predicts attachment representations with their peers (*p*-value = 0.014, well below the significance level of 0.05). The results (R = 0.188, F = 3.080, ddl = 1, and *p*-value = 0.046) indicate that there is a significant linear relationship between exposure to traumatic events and the degree of attachment representations with parents. As a result, the degree of attachment representations with peers (*p*-value = 0.046, well below the 0.05 significance level) predicted exposure to traumatic events. However, the strength of the relationship (R-squared = 0.03) between exposure to traumatic events and the degree of attachment representations with parents is weak. Similarly, exposure to traumatic events has a very weak effect (adjusted R-squared = 0.03) on this relationship (Table 2).

## 4. Discussion

This study was to determine the relationship between trauma experience and the degree of attachment representations in school-aged children in Burundi. To achieve this objective, we described the prevalence of traumatic events likely to have been experienced by the children during their lifetime and the levels of attachment representations of children with their parents and peers. Finally, we examined the relationship between these traumatic experiences and their levels of attachment representations in Burundian school-aged children. The results indicate that 72.30% of the children who experienced or witnessed trauma had higher levels of insecure and disorganized attachment representations with their peers. These same results show that 76.10% of the children had secure attachment representations with their parents. The distribution of the degree of attachment representations for children with their parents is asymptotic and uneven with their peers, regardless of the children’s different living environments. The results of the linear regression test (R = 0.230, F = 6.194, ddl = 1, and *p*-value = 0.014) indicate that exposure to traumatic events is weakly significant to the degree of attachment representations of the children surveyed; it, therefore, follows that attachment representation with their peers (*p*-value = 0.014, well below the significance level of 0.05) predict exposure to traumatic events, with a weight of 0.029 for witnessing a dead body of one’s friend or neighbor. Furthermore, the results (R = 0.188, F = 3.080, ddl = 1, and *p*-value = 0.046) indicate that there is a significant linear relationship between exposure to traumatic events and the degree of attachment representations with parents. As a result, the degree of attachment representations with peers (*p*-value = 0.046) predicts exposure to traumatic events. Physical abuse (B = 6.450) contributed positively and sexual abuse contributed negatively (B = −8.358) to this relationship. From these results, the children’s attachment representations with parents and with peers predict exposure to the traumatic event.

The model is a right-hand equation: Y = 35.513 − 0.363x for attachment with parents and Y = 27.599 − 0.370x for attachment with peers, respectively. The results of this study demonstrate that parental sexual abuse and physical abuse contribute to the degree of attachment that child victims and witnesses have.

Trauma poses a significant burden of risk for the emergence of physical and mental health issues [23]. The exposure of young children to early traumatic experiences can have serious repercussions on their development [24,25]. The traumatisms caused by psychological, physical, or sexual abuse in a setting of contact generally differ greatly from those brought on by disasters, health issues, or conflicts [26]. Thus, the relationship between trauma and attachment is evidenced by Bowlby’s attachment theory [27]. It is a closely related relationship. As *The Study of Bowlby’s Attachment Theory* (1944, 1951, 1960, 1969/1980, 1973, and 1980) explained, exposure to interpersonal trauma such as separation, loss, abuse, or violence has an impact on mental health [27]. Attachment styles have been associated with childhood trauma including rape, kidnapping, and gunshot wounds [28]. Thus, survivors of childhood sexual abuse have a preoccupied and insecure level of attachment [29]. In larger-scale samples, studies have found an association between the events of sexual abuse, physical, or emotional abuse [30], and the type of insecure attachment exhibited by adults [31].

Previous studies [32,33,34,35] that have empirically linked childhood trauma and attachment appear to be consistent with the hypothesis that insecure attachment provides the relationship between childhood trauma and the increased somatic symptoms seen in adults. This hypothesis suggests that childhood trauma causes the development of an insecure attachment that involves the expectation that other people will not meet the individual’s emotional needs. Since Bowlby’s detailed studies in the 1950s and 1960s, the transmission of attachment patterns from parent to child has been a widely debated area of research in child psychology. Studies of identical twins showed that whether they were adoptive or biological twins, it was the mother’s type of attachment throughout the individual’s childhood that was the key factor regarding the child’s attachment style, rather than genetics. Other studies [33,36,37,38,39,40,41,42] have found that insecure attachment types may be partly the result of a poor relationship between the caregiver and their partner.

In early childhood, there appears to be a link between childhood maltreatment and insecure attachment type. Infants exposed to maltreatment tend to have an insecure attachment type [39]. Thus, the emotional abuse experienced in childhood is a significant sign of insecure attachment in adulthood, so it would be more illuminating to consider this than other widely studied forms of abuse such as sexual or physical abuse. The self-trauma model presented by Briere [43] demonstrates how trauma interferes with a child’s development, particularly with their attachment system. The attachment system is one of the fundamental action systems that regulate responses to threats and prevents their interference with everyday action systems. While researchers interested in attachment agree on the importance of trauma on attachment and vice versa, they have gone further with the concept of “attachment-related trauma”. Attachment-related trauma arises from events where “a frightening experience is accompanied by or results from the appraisal of loss, rejection, or abandonment by an attachment figure”.

Although not considered an actual threat to survival, attachment-related trauma in adulthood is detrimental, particularly because the threat to self is accompanied by the threat of loss or abandonment by an attachment figure. Kobak, Cassidy, and Zir [44] propose four types of attachment trauma. The first type of attachment trauma is attachment disruption, or unexpected and/or prolonged separations involving very little communication and no shared plan for a reunion. The second type of trauma involves events where the child is sexually abused by the attachment figure, which is particularly damaging because it generates a dilemma for the child. Physical abuse is related to both attachment styles separately. Anxious-ambivalent and anxious-avoidant attachment styles are similarly related to regulation deficits. Separately, the anxious-ambivalent and anxious-avoidant attachment styles mediate physical abuse and regulatory deficits. 

## 5. Limitations of This Study

This study has certain limitations. First, the focus was limited in that it was a descriptive study. Although this is a challenge, future studies are needed to examine the psychological mechanism that determines the individual’s mental representations of self, others, and attachment patterns in intimate and personal relationships (parents and partners) from a longitudinal perspective. The second limitation is that only self-report measures were used in the data collection. Although participants were asked to be as honest as possible in answering the surveys, it was not possible to verify the extent to which they provided accurate responses to the questions, particularly those that asked for answers regarding the abuse and neglect they were exposed to as children. The third limitation is that all participants were children attending schools in the same area of Maranvya Mairie of Bujumbura in Burundi. In this regard, it would be difficult to generalize the results of this study to samples other than these students. Despite these limitations, the results of this study make some contributions to the current literature in certain aspects. First, in this study, there is a link between traumatic experiences and the degree of attachment representation in students. Second, these results highlight some evidence that traumatic experiences are related to a significant level of child attachment with parents and peers.

## 6. Conclusions

In this study, the children experienced at least one traumatic event in their lifetime. Children who experienced traumatic events have higher levels of insecure and disorganized attachment representations with their peers more frequently than secure attachment representations with their peers. Children’s attachment representations with parents are more secure than insecure. First, in this study, the results highlight some evidence that traumatic experiences are linked to a significant level of children’s attachment to their parents and peers. Traumatic childhood experiences have repercussions on the representations of the attachments of children with their parents and their peers. Children who have higher levels of representations of insecure and disorganized attachment with their peers are more prone to self-harm than those with representations of secure attachment with their peers. Children’s attachment representations with parents are more secure than uncertain. Despite the weakness of the linear relationship, the exposure to traumatic events in school-aged children explains why their level of attachment representations with their parents is low.

## Figures and Tables

**Figure 1 brainsci-13-00666-f001:**
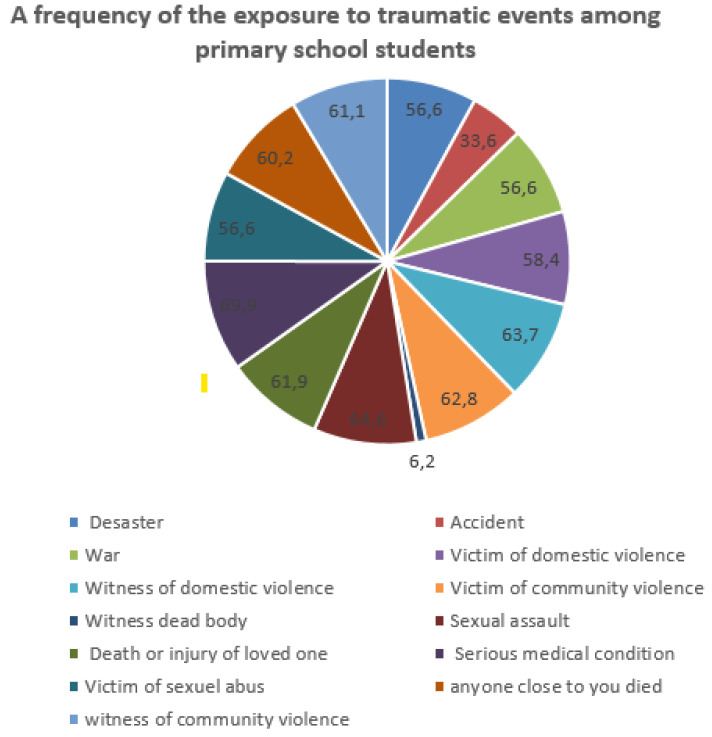
Rate of exposure to traumatic events among primary school students.

**Figure 2 brainsci-13-00666-f002:**
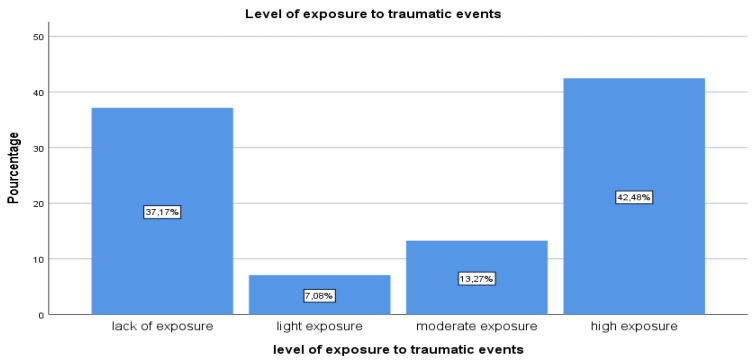
Level of exposure to traumatic events.

**Figure 3 brainsci-13-00666-f003:**
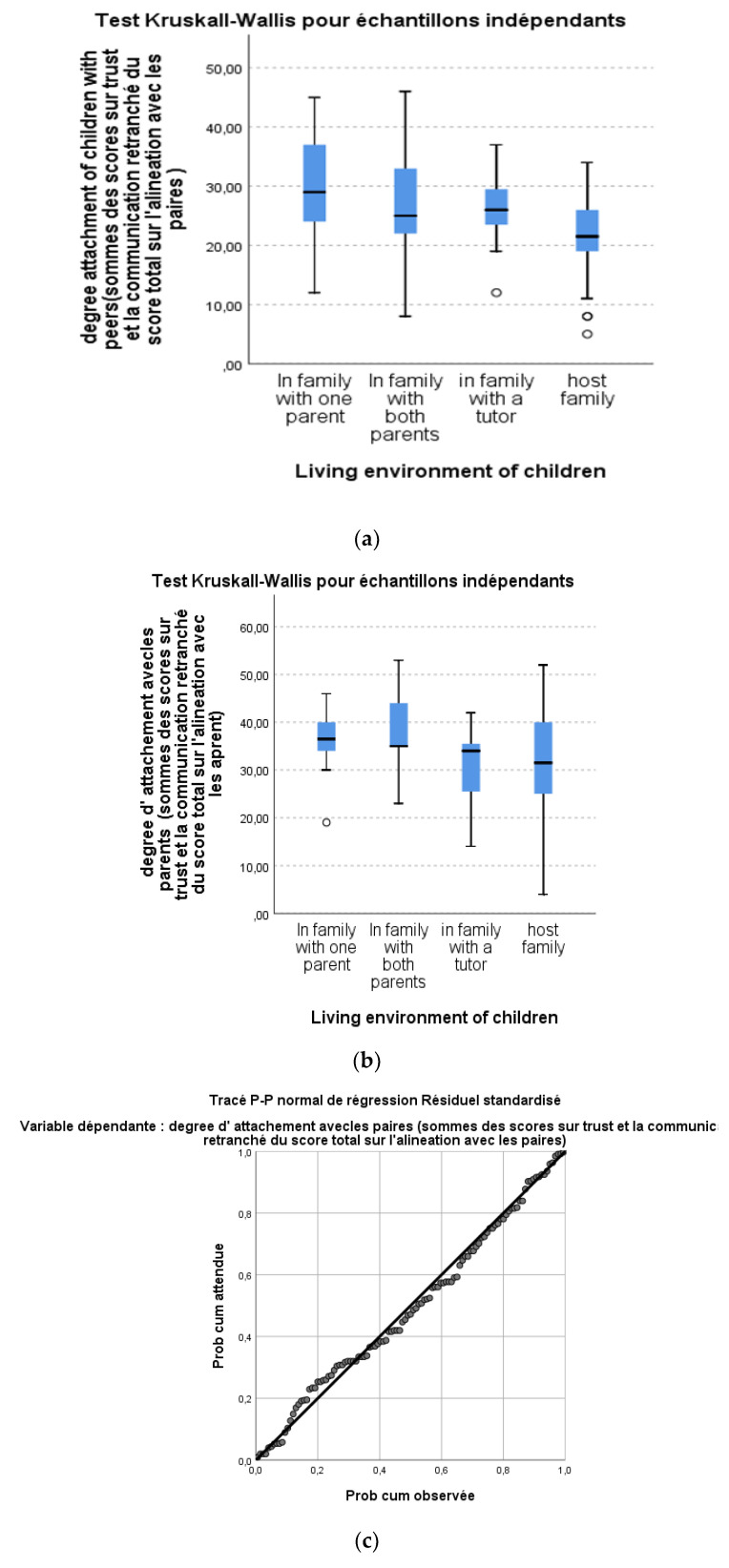
The *Kruskall–Wallis boxplot* for the degree of attachment representations with parents and their peers. (**a**) The distribution of the degree of attachment representations for children with their parents. (**b**) The distribution of the degree of attachment representations for children with their peers. (**c**) Normal *p*-*p* trace of standardized residual regression.

**Table 1 brainsci-13-00666-t001:** Representations of children’s degree of attachment to their peers and parents.

Attachment with Their Peers (*N* = 113)		Attachment with Their Parents (*N* = 113)
Insecure attachments less than 28 (72.30%)	73	27 (23.89%)
Secure attachment greater than (27.69%) than or equal to 28	40	86 (76.10%)

**Table 2 brainsci-13-00666-t002:** Regression model analysis.

Model Sum of Squares	Mean Square	R	R-Squ	β	ddl	F	Sig.
attachment with parent Regression	352.629	0.230	0.07	27.5	1	6.194	0.014
attachment with parents Regression	339.499	0.188	0.03	35.913	1	4.08	0.046

R = multiple correlation coefficient; R-Squ = R-square (determination coefficient); β = beta (standardized coefficient); ddl = degree of freedom; F = Ficher test; sig. = significance level.

## Data Availability

Not applicable.

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
