# Peer review of "The Relationship between Trauma and Attachment in Burundi’s School-Aged Children"

_brainsci, 2023, doi:10.3390/brainsci13040666_

Round 1

Reviewer 1 Report

Comments and Suggestions for Authors

In section 2.2. "participants" is written in capital letters "MARAMVYA COMMUNE MUTIMBUZI" please correct this.

It is necessary to include in this section the sample calculation carried out or otherwise include the statistical power with the data that have been used.

Please define more clearly the inclusion and exclusion criteria.

In section 2.4 "data analysis" please indicate what tests were performed to determine whether or not the data followed a parametric distribution. The results of these tests should be indicated and the distribution of the data should be confirmed.

In section 2.5, the following sub-sections are wrong, it should be 2.5.1, not 2.4.1. The cultural adaptation of the instruments should be detailed in detail, indicating not only the generic procedure but the particular process that has been carried out with the concrete data of what happened.

Some of the instruments have not been validated in the population in which they are administered, so it is necessary to report confirmatory factor analysis data on them to ensure that they work adequately in this population.

The "2.5. Ethical precautions" should be "2.4. Ethical precautions", I think this study should be approved by a scientific committee. I believe that the appropriate persons to sign the informed consent of minors are the parents or legal guardians of the minor.

Resultats" should be "3. Results". Would it be interesting to analyze the results according to sex?

I think that Figure 1 would look better with a bar graph since in the graph used it seems that the total area is 100% and can lead to misinterpretations.

There is information in the results that should be included in statistical analysis, please revise the section and include the information in the appropriate place. Please include the square r's for a correct interpretation of the regression model used.

The conclusion should include the specific conclusions of the study based on the results, not only the lines for the future. In fact, I think it would be more appropriate to include the future lines in the discussion rather than in the conclusion.

Reviewer 2 Report

Comments and Suggestions for Authors

The article needs a lot of corrections/improvements to meet the journal's publication standards, and I will be as accurate as possible:

1.      The authors did not follow the journal template.

2.      Please enter bibliographic references as required by the journal: Author 1, A.B.; Author 2, C.D. Title of the article. Abbreviated Journal Name YearVolume, page range.

3.      Reading the article's abstract, I noticed that the authors started directly with the study's objective.  Please insert the background, place the question addressed in a broad context, and highlight the purpose of the study.

4.      At the end of the introduction, this study's novelty elements and the paper's purpose must be included.

5.      In sub-chapter 2.2.  Participants, please specify more clearly what the inclusion criteria were.

6.      The data were analyzed using the statistical software SPSS (version 22 of Windows; SPSS)[16].  Please follow the model: IBM SPSS 23.0 Statistics (IBM Corp. Released 2015.  IBM SPSS Statistics for Windows, Version 23.0.  IBM Corp., Armonk, NY, USA).

7.      Please enter the sample size calculation or statistical power.

8.      The original version of the People in My Life questionnaire [20].  The original version of attachment to parents and peers: People in My Life (Cook, Greenberg, & Kusche, 1995 https://www.researchgate.net/publication/231967046_Promoting_Emotional_Competence_in_School-Aged_Children_The_Effects_of_the_PATHS_Curriculum  ) and is a downward extension of Armsden and Greenberg's (1987) Inventory of Parent and Peer Attachment, which was initially developed through factor analysis with a college student sample to tap behavioural elements of adolescent attachment and affectively toned cognitive expectancies, suggestive of the internal working models of attachment to parents and close friends:

·        https://link.springer.com/article/10.1007/BF02202939

·        https://www.researchgate.net/publication/258923895_The_Inventory_of_Parent_and_Peer_Attachment_Individual_Differences_and_Their_Relationship_to_Psychological_Well-Being_in_Adolescence

9.      In the discussion part, please start with the study's objectives, not the study's purpose.

10.   Previous studies have empirically linked childhood trauma and attachment appear to be consistent with the hypothesis that insecure attachment provides the relationship between childhood trauma and increased somatic symptoms seen in adults.  Please kindly cite those studies.

11.   Other studies have found that insecure attachment types may be partly the result of a poor relationship between the caregiver and the partner [19].  Please kindly cite those studies.

12.   Kobak, Cassidy, and Zir (32) propose four types of attachment trauma.  The bibliography does not contain reference 32.

13.   This study has certain limitations.  Please insert 5.  Limitations of this study.

14.   Conclusions should be brief and to the point, based on the study's results.

15.   The article was included to analyze for the similarity coefficient in the Plagiarism CheckerX software, version 6.0.11, and please perform the following word reformulations that are in bold and italics:

·        Children with secure maternal and paternal attachments perceived their parents as less rejecting, while children with secure paternal attachments also reported their parents to be emotionally warmer [10].

·        Mental, physical, or sexual abuse in close personal relationships generally results in trauma that is very different from that caused by accidents, illness, or war [22].

Round 2

Reviewer 1 Report

Comments and Suggestions for Authors

The authors have responded to all my comments, I have no additional comments.

Reviewer 2 Report

Comments and Suggestions for Authors

1.  Please leave the statistical programme part this way: IBM SPSS 26.0 Statistics (IBM Corp. Released 2019. IBM SPSS Statistics for Windows, Version 26.0. Armonk, NY: IBM Corp).

2.  Please replace 3. Resultats with 3. Results